# Long-Term Lithium Therapy and Thyroid Disorders in Bipolar Disorder: A Historical Cohort Study

**DOI:** 10.3390/brainsci13010133

**Published:** 2023-01-12

**Authors:** Boney Joseph, Nicolas A. Nunez, Vanessa Pazdernik, Rakesh Kumar, Mehak Pahwa, Mete Ercis, Aysegul Ozerdem, Alfredo B. Cuellar-Barboza, Francisco Romo-Nava, Susan L. McElroy, Brandon J. Coombes, Joanna M. Biernacka, Marius N. Stan, Mark A. Frye, Balwinder Singh

**Affiliations:** 1Department of Psychiatry & Psychology, Mayo Clinic, Rochester, MN 55905, USA; 2Department of Neurology, Mayo Clinic, Rochester, MN 55905, USA; 3Department of Quantitative Health Sciences, Mayo Clinic, Rochester, MN 55905, USA; 4Department of Psychiatry, Universidad Autónoma de Nuevo León, Monterrey 64460, Mexico; 5Lindner Center of HOPE, Department of Psychiatry and Behavioral Neurosciences, University of Cincinnati, Cincinnati, OH 45040, USA; 6Division of Endocrinology, Diabetes, Metabolism, and Nutrition, Mayo Clinic, Rochester, MN 55905, USA

**Keywords:** lithium, bipolar disorder, thyroid, mood disorders, retrospective studies

## Abstract

Lithium has been a cornerstone treatment for bipolar disorder (BD). Despite descriptions in the literature regarding associations between long-term lithium therapy (LTLT) and development of a thyroid disorder (overt/subclinical hypo/hyperthyroidism, thyroid nodule, and goiter) in BD, factors such as time to onset of thyroid abnormalities and impact on clinical outcomes in the course of illness have not been fully characterized. In this study we aimed to compare clinical characteristics of adult BD patients with and without thyroid disorders who were on LTLT. We aimed to identify the incidence of thyroid disorders in patients with BD on LTLT and response to lithium between patients with and without thyroid disorders in BD. The Cox proportional model was used to find the median time to the development of a thyroid disorder. Our results showed that up to 32% of patients with BD on LTLT developed a thyroid disorder, of which 79% developed hypothyroidism, which was corrected with thyroid hormone replacement. We did not find significant differences in lithium response between patients with or without thyroid disorders in BD. Findings from this study suggest that patients with BD and comorbid thyroid disorders when adequately treated have a response to lithium similar to patients with BD and no thyroid disorders.

## 1. Introduction

Thyroid functioning plays an important role in mood stabilization, especially in bipolar disorder (BD) [1]. The prevalence of abnormal thyroid function in the US population varies, with reported rates of elevated and decreased TSH levels of 9.5% and 2.2%, respectively [2]. Hypothyroidism, just like mood disorders, is more common in females than males [3,4]. One of the probable causes of thyroid dysfunction in BD is the use of Lithium (Li) therapy [5]. Several studies have identified risk factors for Li-induced hypothyroidism such as the thyroid peroxidase antibodies, female sex, older age, and a positive family history of hypothyroidism [6,7,8]. Thyroid autoimmunity, a common cause for hypothyroidism, is prevalent in BD, which may or may not be associated with Li treatment [9]. Hypothyroidism can contribute to rapid cycling in BD [10], although this is not shown consistently [11]. Li may also cause hyperthyroidism and rarely Graves’ disease [12,13]. Lithium is a gold standard mood stabilizer for BD; however, it continues to remain underutilized, to a greater extent in the US [14,15,16]. Thyroid and renal dysfunctions associated with probable Li use, requiring regular monitoring of renal and thyroid indices, have been proposed as some of the reasons for underutilization of Li [15,17,18]. However, whether Li-induced hypothyroidism warrants the discontinuation of Li remains to be elucidated.

Pre-clinical studies have shown that Li has several biological effects on the thyroid gland, some of which include an increase in iodine content within the thyroid gland; reduction in thyroid gland activity for production of thyroxine (T4) and triiodothyronine (T3); blocking the release of thyroid hormones from the gland; and alteration in the thyroglobulin structure (a protein in the thyroid gland that is essential in the formation of thyroid hormones). Due to its action on the thyroid gland, Li can be used as an augmentation agent for the treatment of hyperthyroidism [12].

While there is consensus among the clinical practice guidelines for the treatment of overt hypothyroidism with thyroid hormone replacement (THR), recommendations for management of subclinical hypothyroidism (SCH—elevated TSH and normal free T4) remain inconsistent. Recent clinical practice guidelines on SCH recommended THR when patients are younger, planning pregnancy, symptomatic, or have cardiovascular disease or dyslipidemia [19]. The same clinical practice guidelines reported no significant change in depressive symptoms with THR. Thyroid disorders—both hypo- and hyperthyroidism—can create a clinical conundrum for treating clinicians in managing BD patients who are on long-term lithium therapy (LTLT).

Considering the dearth of real-world clinical practice data to guide clinicians regarding optimal decision choices in clinical practices following development of thyroid disorders in BD patients receiving LTLT, we conducted a historical cohort study and investigated the association between LTLT and the development of thyroid disorders (overt/subclinical hypo/hyperthyroidism, thyroid nodule, and goiter). We investigated the Li response in patients who developed thyroid disorders compared to those who did not, as well as the time to development of thyroid disorders. Our secondary aim was to investigate the difference in the time to thyroid disorder development between males and females.

## 2. Methods

In this historical cohort study, we included adult patients (≥18 years) with BD on LTLT (i.e., lithium prescribed for a minimum of 1 year) who were enrolled in the Mayo Clinic Bipolar Biobank at Mayo Clinic in Rochester, Minnesota [16,20], from 1 July 2009 to 5 May 2018. All included patients fulfilled the DSM-IV-TR criteria and diagnostic confirmation of BD-type I/BD-type II or schizoaffective bipolar subtype [21]. Our exclusion criteria were patients not on LTLT. The Mayo Clinic Bipolar Biobank was created in collaboration with the Linder Center of HOPE/University of Cincinnati and the University of Minnesota to promote and foster research on disease risk and pharmacogenomic treatment response in BD [20]. This study was approved by the Mayo Clinic Institutional Review Board.

The data on demographics, medical comorbidities [22], duration of Li treatment, and relevant clinical characteristics were extracted from the electronic medical records (EMR). Laboratory data were extracted using the Mayo Data Explorer, available at the Mayo Clinic. Duration of Li therapy was calculated from chart review by adding the duration of different Li trials as published earlier [16,23]. To assess the medical illness burden, we used the Modified Cumulative Illness Rating Scale (MCIRS) [24]. The MCIRS identifies 14 items involving different systems with a score range from 0 to 4. The severity index is the mean of the scores of the first 13 categories—excluding psychiatric comorbidity—to study the comorbidity burden; individual category mean score was evaluated to study the specific comorbidity.

Treatment response to Li was assessed using the Alda-A scale [25]. Alda-A scale is a composite measure of clinical improvement in severity, duration, and frequency of illness and is rated from 0 to 10. Response to Li treatment was defined as good, moderate, or poor response based on Alda-A score ≥ 7, 4–6, and ≤3, respectively. Internal consistency was previously assessed with Kappa statistics (intraclass correlation coefficients) for Alda-A scores between raters reaching 0.968, 95% CI (0.933, 0.986) [26]. We elected to use the A subscale since it has shown to have comparable intraclass correlation compared to the total score [27]. However, the Alda-A score was not corrected for any confounding factor.

### 2.1. Ascertainment of the Thyroid Disorder

Diagnosis of thyroid disorders was adjudicated via clinical diagnoses in the EMR (as documented by treating clinicians) or according to the TSH and free T4 levels. Thyroid disorders were divided into the following functional subtypes: overt hypothyroidism, subclinical hypothyroidism, overt hyperthyroidism, and subclinical hyperthyroidism. From a structural perspective, we identified benign nodule, benign diffuse goiter, toxic nodule, and toxic multinodular goiter. When the etiology was known, the diagnosis of Hashimoto’s thyroiditis or Graves’ disease was also listed.

### 2.2. Outcomes

Our primary outcome was the development of incident thyroid disorders. We investigated the difference in the time to thyroid disorder development between males and females. Our secondary outcome was response to Li between patients with and without thyroid disorders. We reported data on the incident thyroid disorder cases among BD patients on LTLT. We also compared TSH levels at baseline, and by the end of the study between patients with and without thyroid disorders.

### 2.3. Statistical Analysis

Descriptive characteristics for categorical variables were summarized as frequencies and significance differences were tested using chi-square test and Fisher’s exact test depending on the number of elements in each cell. Continuous variables were reported as mean (standard deviation (SD), median (interquartile range (IQR) and categorical variables as counts and percentages. The unpaired Student’s-t test and Mann–Whitney U test were used to compare continuous variables with normal distribution and for skewed distribution, respectively.

For analyses with thyroid disorders as the outcome, patients were studied from the start date of LTLT until the occurrence of thyroid disorders, the study end date, or the last date available in the EMR, whichever came first. The Cox proportional hazards model was used to estimate the hazard ratio and median time to the development of thyroid disorders between males and females after starting Li. Kaplan–Meier plots were used to show the time from Li initiation to the development of thyroid disorders between males and females. All analyses were performed using R software for statistical computing (version 4.2.2) in RStudio IDE (version 2022.12.0+353) [28,29]. *p*-values ≤ 0.05 were considered significant.

## 3. Results

The study included 154 BD patients on LTLT. At baseline, 16 patients had preexisting thyroid disorders (prevalent cases) before starting Li and 2 patients had missing thyroid status at baseline and thus were excluded from the primary analysis. During LTLT, 43 patients (31.6%) developed incident thyroid disorders. Clinical characteristics of the study population (N = 136) are described in Table 1. Participants were predominantly females (57.4%) with BD-I (BD-I/II distribution of 69.9%/27.9%, respectively), median age of 40.5 years (at study enrollment), with a median time to follow-up of 7.9 years. Compared to the non-thyroid disorder group, patients with thyroid disorders had higher median MCIRS scores (18 vs. 21; *p* = 0.01) and borderline increased history of trauma (32% vs. 50%, *p* = 0.05). There were no other significant differences in terms of demographic and clinical variables between the thyroid disorders and non-thyroid disorders groups.

At baseline, 16 patients had prevalent thyroid disorders and a total of 61 patients had thyroid disorders by the study end date. At one year of Li treatment, 7% (n = 9/131) of patients developed thyroid disorders. Thirty-two percent of patients (n = 43/136) developed new thyroid disorders by the study end date, of which 79% (n = 34/43) developed overt or subclinical hypothyroidism and 94% (32/34) of these patients (with subclinical/overt hypothyroidism) were on THR (Table 2). The majority of the patients who were on THR received treatment with levothyroxine. Regarding hyperthyroidism, one patient was on methimazole and propranolol, respectively. Only two patients discontinued Li due to the development of thyroid disorders.

The median time to the development of a thyroid disorder was significantly lower in females compared to males (17.0 vs. 42.6 years, *p* = 0.04), Figure 1. Compared to males, females had a higher risk of thyroid disorder (HR = 2.00, 95% CI, 1.02 to 3.95: log-rank *p* = 0.04).

In Figure 2, we present the frequency distribution of diagnosis of new thyroid abnormality over the course of Li treatment. In our study, 7% of patients developed thyroid disorders within one year of lithium therapy and 60.5% of patients developed thyroid disorders within five years of lithium therapy. More than one-third of patients developed thyroid disorders after five years of lithium therapy.

Table 3 shows the Li response (measured with Alda-A score) and laboratory values between the thyroid disorder and non-thyroid disorder groups. There were no significant differences in the Li response between patients with or without thyroid disorders. The mean Alda-A scores were also similar between the thyroid disorder and non-thyroid disorder groups (6.26 vs. 6.38; *p* = 0.92). BD patients with thyroid disorders had similar TSH levels at baseline compared to the non-thyroid disorder group (2.26 vs. 1.76, *p* = 0.06), but the value was significantly higher at 1 year. There were no other significant differences between the two groups.

## 4. Discussion

In this historical cohort study, we investigated the effects of LTLT on thyroid function among patients with BD. About a third of patients with BD on LTLT developed a thyroid disorder during the evaluation period, the majority of whom (79%) developed hypothyroidism (subclinical or overt). These findings are similar to a recent study by Lieber and colleagues, where they observed 27% of patients with BD developed overt hypothyroidism [30]. In our cohort, the majority of the patients with hypothyroidism were treated with THR and only two patients with thyroid disorders discontinued Li. The prevalence of thyroid disorders approximates up to 10% in the general population [2] and increases up to 20% in patients with BD [31]. Specifically, studies have underscored that 33% of patients with BD and mixed affective states had TSH abnormalities and 7% in cases of pure manic states [32]. Notably, treatment with Li has been reported to increase prevalence rates of thyroid disorders to up to 54% [7].

In our study, 7% of patients developed thyroid disorders within one year of lithium therapy. This is lower than reported in some of the earlier studies [33]. A Swedish retrospective cohort study reported a mean delay from lithium to initiation of THR of 2.3 years (SD 4.7) with a median of 10 months [30]. A plausible explanation for these findings may relate to their inclusion of patients with BD and schizoaffective disorders in the depressive phase and focusing on those patients who discontinued Li. Our study showed a lifetime prevalence of thyroid disorder diagnosis in approximately 40% of the study cohort, with approximately 32% incident cases of thyroid disorder diagnosis by the end of the study. These findings are in accordance with the broader literature, which underscores higher rates of thyroid disorder with prevalence rates up to 47% in all Li-treated patients [12,34]. Reassuringly, we found no significant differences in Li response between patients with or without thyroid disorders when adequately treated, which further emphasizes the importance of properly treating thyroid dysfunction while continuing Li therapy to maintain mood stabilization.

We found that females had a shorter time to the development of a thyroid disorder compared to males and had double the risk (HR = 2.00) of developing thyroid disorders compared to males. These findings also align with previous studies that have highlighted higher rates of hypothyroidism associated with females [5]. Our results showed higher MCIRS scores in the thyroid disorder group compared to the non-thyroid disorder group, highlighting a higher comorbidity burden in BD with the thyroid disorder group. Autoimmune disorders and family history of thyroid abnormalities are additional risk factors for Li-induced hypothyroidism [35,36]. BD has been reported to be associated with immune dysregulation and autoimmune disorders [37,38]. The role of immunological and immuno inflammatory hypothesis both in the pathophysiology of the disease and treatment side-effects remains an area of active investigation [39].

Lastly, our results underscore that BD patients with thyroid disorders had a borderline higher prevalence of history of psychological trauma compared to the non-thyroid disorder group. History of trauma has been associated with alterations in the HPT axis and immune dysfunction, leading to increased risk of thyroid and autoimmune disorders [40,41], which prompts further exploration of the pathophysiology and interconnection between thyroid-bipolar illness and history of psychological trauma.

### Clinical Implications

Patients with BD receiving LTLT, especially females, are at a higher risk of developing thyroid disorders. This is consistent with prior studies; thus, it is advisable to have a close monitoring of thyroid functioning, especially in females [8]. Findings from this study suggest that patients with BD and comorbid thyroid disorders when adequately treated with THR have a response to Li similar to patients with BD and without thyroid disorders. Further, THR has been used as augmentation strategies for resistant depression [42,43]. These results are reassuring and provide evidence to support continuing Li therapy in patients with thyroid disorders and highlights the importance of collaboration with patients’ primary care providers/endocrinologists if needed. Given that many cases developed after the first year of lithium therapy, it is advisable to continue periodic monitoring of thyroid function long term.

## 5. Strengths and Limitations

Our study has several strengths. We performed a thorough and comprehensive review of clinical data and used robust statistical methods to explore their correlations. Our study cohort includes patients from the Bipolar Biobank, which closely mirrors a real-world clinic representation of BD patients treated with LTLT. Our study cohort also has a longer median follow-up time, which has helped us measure and compare the longitudinal and time-varying effects of LTLT on the thyroid.

While interpreting these findings, we should also be wary of its several limitations. Apart from the inherent shortcomings of a retrospective study design including confounding biases, missing data, and lack of homogeneity across the collected data, our study is also limited by its smaller cohort size and its limited diversity. For our results to be generalized, the study needs to be repeated in larger and more diverse populations in a prospective setting. We also did not explore in detail any of the intrinsic or extrinsic factors that may have predisposed a patient toward a state of immune dysregulation and thyroid abnormality. Our study results thus lack the ability to infer causality, and it describes the association of thyroid disorder only in long-term lithium treated patients with BD.

## 6. Conclusions

One-third of patients with BD on LTLT developed thyroid dysfunction, with similar Li response rates between patients with thyroid disorders and non-thyroid disorders. Most patients with thyroid disorder were on thyroid replacement therapy. The high rates of thyroid disorders shown in this study associated with LTLT complement previous literature and suggest, as clinical guidelines, a recommendation to closely monitoring thyroid hormone function during Li treatment for early identification and diagnosis of thyroid disorders.

## Figures and Tables

**Figure 1 brainsci-13-00133-f001:**
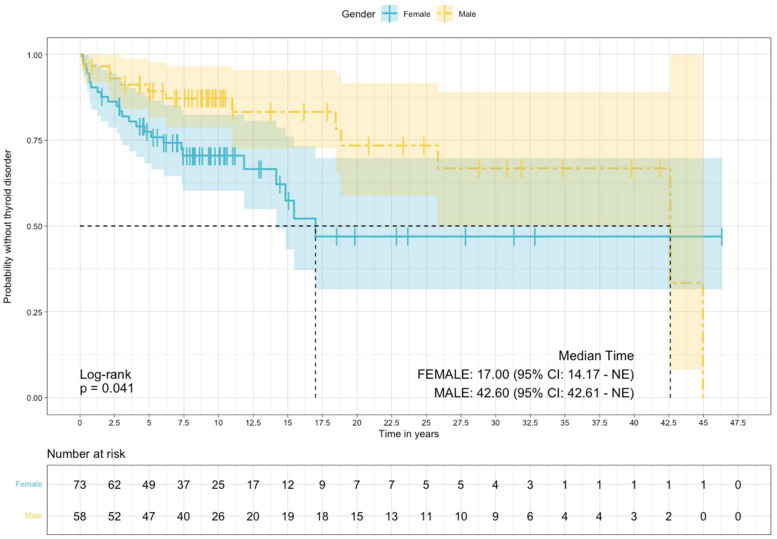
Survival, free of thyroid disorders estimated via the Kaplan–Meier method, stratified by sex.

**Figure 2 brainsci-13-00133-f002:**
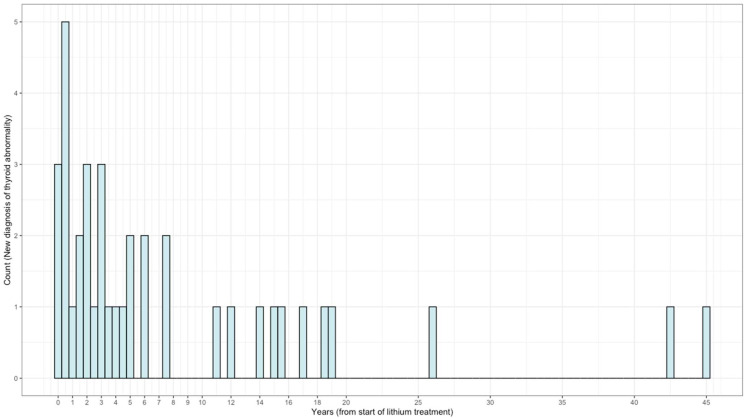
Frequency distribution of diagnosis of new thyroid abnormality during lithium treatment (N = 38).

**Table 1 brainsci-13-00133-t001:** Study population characteristics, reported as Median (Q1, Q3) or N (%).

	All (N = 136)	No Thyroid Disorder (n = 93)	Thyroid Disorder (n = 43)	*p*-Value
Age at enrollment	40.5 (29.7, 55.0)	40.3 (29.4, 54.8)	40.7 (32.2, 55.8)	*p* = 0.64
Age at study end	47.5 (37.0, 60.0)	47 (35.0, 60.0)	48 (37.5, 60.5)	*p* = 0.72
Gender (Female)	78 (57.35)	48 (51.61)	30 (69.77)	*p* = 0.06
Race (Non-White)	10 (7.35)	5 (5.38)	5 (11.63)	*p* = 0.29
BMI (n = 134)	29.5 (25.1, 35.5)	29.3 (25.4, 35.4)	24.6 (25.0, 35.8)	*p* = 0.93
Diagnosis				
BD-I	95 (69.85)	65 (69.89)	30 (69.77)	
BD-II	38 (27.94)	25 (26.88)	13 (30.23)	
BD NOS/SZ	3 (2.21)	3 (3.23)	0 (0.00)	*p* = 0.64
Age at BD diagnosis (n = 133)	27 (21.0, 41.0)	29 (22.0, 42.0)	25 (20.2, 36.2)	*p* = 0.42
Age at first depressive episode (n = 108)	22 (16.0, 34.0)	23 (16.2, 34.0)	21 (16.2, 32.0)	*p* = 0.82
Age at Lithium start (n = 135)	32.4 (24.0, 44.9)	33.6 (24.1, 45.6)	32 (23.7, 40.2)	*p* = 0.51
Lithium treatment duration in years (n = 135)	8.84 (5.51, 14.9)	8.73 (5.45, 13.1)	11 (6.05, 17.3)	*p* = 0.37
Married (n = 130)	67 (51.54)	45 (51.14)	22 (52.38)	*p* = 1.00
Education ≥ 12 years (n = 132)	129 (97.73)	88 (96.70)	41 (100.00)	*p* = 0.55
History of ADHD	20 (14.71)	16 (17.20)	4 (9.30)	*p* = 0.30
History of Psychosis (n = 133)	62 (46.62)	42 (46.15)	20 (47.62)	*p* = 1.00
History of Trauma (n = 133)	50 (37.59)	29 (31.87)	21 (50.00)	*p* = 0.05
Suicide attempts ≥ 1 (n = 134)	38 (28.36)	26 (28.26)	12 (28.57)	*p* = 1.00
Alcohol Use Disorder (n = 133)	46 (34.59)	28 (30.77)	18 (42.86)	*p* = 0.33
Family History (First-degree)				
Bipolar Disorder	35 (25.74)	23 (24.73)	12 (27.91)	*p* = 0.68
Major Depression	50 (36.76)	35 (37.63)	15 (34.88)	*p* = 0.85
Substance abuse	37 (27.21)	24 (25.81)	13 (30.23)	*p* = 0.68
MCIRS score	19 (15.8, 23)	18 (15, 22)	21(18, 26)	*p* = 0.01

Abbreviations: BD-I = Bipolar type I; BD-II = Bipolar type II; BD NOS = Bipolar disorder not otherwise specified; ADHD: Attention-Deficit/Hyperactivity Disorder; SZ = Schizoaffective disorder; MCIRS = Modified Cumulative Illness Rating Scale.

**Table 2 brainsci-13-00133-t002:** Thyroid status of patients with abnormal thyroid functioning.

	Baseline	Study End	On Thyroid Replacement at Study End ^a^
	(Before Lithium)	(New/Excluding baseline)	(New/Excluding baseline)
Hypothyroid /Hashimoto’s	11 (5)	29 (0)	29 (0) ^b^
Subclinical Hypothyroid	2	5	3
Hyperthyroid (Graves’ disease)	2 (2)	0 (0)	0
Subclinical Hyperthyroid	0	1	0
Benign Nodule/Goiter	1	7	3 ^c^
Toxic Nodule/Goiter	0	1	1 ^d^
Total	16	43	36
Data not available	2	0	0

^a^ Thyroid replacement/augmentation was with T4 for all patients; ^b^ One patient stopped Li following thyroid dysfunction, but continued thyroid replacement; ^c^ One patient on T4 augmentation, two patients on replacement following thyroidectomy; ^d^ One patient on thyroid replacement following thyroidectomy.

**Table 3 brainsci-13-00133-t003:** ALDA Response and Laboratory values.

	**All** **(N = 136)**	**No Thyroid Disorder (n = 93)**	**Thyroid Disorder** **(n = 43)**	** *p* ** **-Value**
ALDA (n = 123)				
ALDA Responders, n (%)	61 (49.59)	42 (49.41)	19 (50.00)	*p* = 1.00
ALDA—A Score, Mean (SD)	6.34 (2.35)	6.38 (2.27)	6.26 (2.54)	*p* = 0.92
ALDA Response, n (%)				*p* = 0.85
Good	61 (49.59)	42 (49.41)	19 (50.00)	
Moderate	48 (39.02)	34 (40)	14 (36.84)	
Poor	14 (11.38)	9 (10.59)	5 (13.16)	
Lab (Mean (SD)				
S. TSH Baseline (n = 77)	1.92 (1.03)	1.76 (0.95)	2.26 (1.15)	*p* = 0.06
S. Lithium After 1 year Li (n = 77)	0.69 (0.28)	0.70 (0.28)	0.65 (0.27)	*p* = 0.32
S. TSH After 1 year Li (n = 65)	3.17 (2.19)	2.66 (1.3)	4.30 (3.20)	*p* = 0.05
S. Lithium End of study (n = 71)	0.69 (0.41)	0.63 (0.39)	0.78 (0.43)	*p* = 0.13
S. TSH End of study (n = 73)	2.42 (1.43)	2.34 (1.14)	2.55 (1.80)	*p* = 0.96

## Data Availability

The data are not publicly available because of restrictions; they contain information that could compromise the privacy of the research participant. The data that support the findings of this study are available on reasonable request from the corresponding author.

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
