# Peer review of "Long-Term Lithium Therapy and Thyroid Disorders in Bipolar Disorder: A Historical Cohort Study"

_brainsci, 2023, doi:10.3390/brainsci13010133_

Round 1
Reviewer 1 Report
The authors conducted an interesting retrospective study to characterize the incidence of development of thyroid disorders in patients under long-term lithium treatment, evaluate the effect of gender, as well as compare patients who developed or not thyroid disorders in terms of clinical response to lithium.
The study is of great interest as it adds to the literature on adverse effects during lithium treatment. This mood stabilizer is the gold-standard in the long-term treatment of bipolar disorder but it is underutilized, especially in the US, due to different factors among which there is the potential development of adverse effects. Therefore, studies that are able to provide information in a real-life context on these adverse effects and their management during lithium therapy are of high value.
The method are sound, the criteria used to select patients to include are well specified and the conclusion are supported by the results. I only have minor comments.
- I understand the reasoning for using the A score of the Alda scale based on the cited study. However, can the authors elaborate a bit on the fact that this assessment of lithium response is not corrected for any confounding factor?
- At page 8, line 4, the expression “majority of which” should be replaced with “the majority of which”
- In the conclusion, the expression “One-third BD patients” should be replaced with “One-third of BD patients”
Author Response
The method are sound, the criteria used to select patients to include are well specified and the conclusion are supported by the results. I only have minor comments.
- I understand the reasoning for using the A score of the Alda scale based on the cited study. However, can the authors elaborate a bit on the fact that this assessment of lithium response is not corrected for any confounding factor?
Reply. Thank you. We have added a sentence that Alda-A does not correct for any confounding factor.
- At page 8, line 4, the expression “majority of which” should be replaced with “the majority of which”
Reply. Thank you. We have made the suggested change.
- In the conclusion, the expression “One-third BD patients” should be replaced with “One-third of BD patients”
Reply. Thank you. We have made the suggested change.
We have comprehensively reviewed our manuscript for the English language and corrected any grammatical errors.
Reviewer 2 Report
No extra recommendation.
Author Response
Comments and Suggestions for Authors
No extra recommendation.
Reply. Thank you.
Reviewer 3 Report
Dear authors,
Here are my comments referring to your work:
1. I think you could write in an equivalent way the aims - in the abstract, introduction, outcomes and conclusions.
2. In the section Keywords, you could write: bipolar disorder instead bipolar depression.
3. In the table 2 - bold characters for (p=0.05) (history of trauma)
3. Lines 180-181: there aren't explanations about the results from the figure 2.
4. Section Discussion to be more developed.
I would like to thank you for the opportunity to review this article.
Author Response
Thank you for your helpful comments. We have replied to them point-by-point.
1. I think you could write in an equivalent way the aims - in the abstract, introduction, outcomes and conclusions.
Reply: Unfortunately, the journal does not allows us to format the abstract in a structured manner. Else, we would have been happy to make the changes.
2. In the section Keywords, you could write: bipolar disorder instead bipolar depression.
Reply: Thank you. We have made the suggested change.
3. In the table 2 - bold characters for (p=0.05) (history of trauma)
Reply: Thank you. Actually, the p-value is only borderline at 0.0549. That's why we decided not to bold it. To maintain the table's formatting, we consolidated 0.0549 to 0.05.
4. Lines 180-181: there aren't explanations about the results from the figure 2.
Thank you. We have added more information.
"In our study 7% of patients developed thyroid disorders within one year of lithium therapy and 60.5% patients developed thyroid disorders within five years of lithium therapy. More than one-third of patients developed thyroid disorders after five years of lithium therapy."
5. Section Discussion to be more developed.
Thank you. We have made changes to the discussion section.